# Challenges in teacher-student communication during family medicine residency: A qualitative study

**Isaías Hernández-Torres[1], Octavio N. Pons-Álvarez[1], Luisa F. Romero-Henríquez[2]\*, Geovani López-Ortiz[1]\***

1 Facultad de Medicina, Subdivisión de Medicina Familiar, Universidad Nacional Autónoma de México, Ciudad de México, México, 2 Facultad de Filosofía y Letras, Posgrado en Pedagogía, Universidad Nacional Autónoma de México, Ciudad de México, México

\* fernandaromero55@yahoo.com.mx (LFR-H); geovani.lorz@fmposgrado.unam.mx (GL-O)

## Abstract

### Background

Deficiencies in communication among healthcare professionals, recognized by medical educators and healthcare institutions, can negatively impact medical education and clinical practice. Analyzing teacher-resident communication difficulties shed light on this issue and propose practical strategies for its mitigation.

### Objective

To identify common communication challenges between teacher and residents during Family Medicine residency and to analyze their impact on interactions with peers, the work team, and patients.

### Design

Qualitative study, the critical incident technique was used to collect information of interest.

### Participants

Seventy teachers, and fifty third-year residents from the Mexican Republic described critical incidents related to their communication experiences during Family Medicine residency.

### Results

192 critical incidents were collected (several participants reported more than one incident), comprising 127 reports from teachers, and 65 from residents. Four themes were identified: 1) asymmetric communication, 2) assertive communication, 3) organizational communication, and 4) effective communication. The main challenges identified were abuse of power in communication, lack of communication skills, and the absence of institutional communication channels. These issues significantly impacted learning, work environment, interpersonal relationships, and medical care.

**Data Availability Statement:** There are restrictions on sharing the data due to the sensitive nature of the information collected. Participants disclosed communication problems within the family

medicine residency program; many narratives could be easily identifiable. Participants could be academically and professionally affected if they are identified through their narratives, which would compromise the informed consent they signed and the principle of non-maleficence. Therefore, the data can only be shared upon request through the Research Coordination of the Family Medicine Subdivision at the National Autonomous University of Mexico. The contact e-mail is investigacion. smf@fmposgrado.unam.mx.

**Funding:** The author(s) received no specific funding for this work.

**Competing interests:** The authors have declared that no competing interests exist.

## Conclusion

This study highlights communication issues within Family Medicine residency in Mexico. The issues detected hindered learning and effective collaboration and negatively impacted the work environment, interpersonal relationships, and the quality of medical care. These findings underscore the urgent need to reorient the medical specialty curriculum towards an approach that includes communication skills.

## Introduction

Medical residency is pivotal in specialists' training; however, it tends to be asymmetrical towards care aspects, often deprioritizing the development of complementary skills such as communication [1]. Several studies have shown that deficiencies in communication skills among family medicine residents and specialists are a systemic problem caused by inadequate training and time constraints, which negatively impact interactions in academic and service environments [2–6]. Evaluating these skills comprehensively allows for the identification of areas for improvement [4], biases and disparities in interaction environments [5], and the need to implement specific cultural strategies to address these deficiencies [6]. Therefore, the need to incorporate practical activities for teaching effective communication strategies has been identified in various environments where residency takes place [5].

Despite the importance and recognition of developing these skills in different parts of the world, communication problems are common in Mexico, where cultural and educational frameworks have historically emphasized technical and clinical skills over communication abilities. As a result, content related to communication in various family medicine curricula in the country is not sufficiently explored or nonexistent [7–10].

It has been emphasized that medical education should not only focus on creating experts in the diagnosis, and treatment of diseases, but also foster clear, empathetic, and compassionate communication with patients, and their families [11,12]. Effective communication in medical practice has been linked to increased patient satisfaction, therapeutic adherence, and reduced hospital readmissions [13,14]. Conversely, poor communication can lead to conflicts within institutions, impact on trust, damage professional relationships, and compromise the quality of care [15].

Positive interactions between teachers and students promote motivation, engagement, retention, and student well-being, contributing to academic success [16–18]. The role of communication in positive workplace relationships underscores the need to understand how communication impacts various scenarios in physicians' professional development [13,19], and the role played by academic figures in these processes [20].

Given these considerations, it is essential to identify communication challenges between teachers, and residents during Family Medicine residency. This insight can have significant implications for academic training and professional development.

## Methods

### Study design and setting

Multi-center qualitative study, involving Family Medicine professors and residents from the Mexican Republic. The present research was approved by the Research Ethics Committee of the Faculty of Medicine at the National Autonomous University of Mexico (UNAM);

registration number: FM/DI/010/2021 and adhered to the SRQR guidelines [21]. The study focused on exploring teacher-resident communication challenges–which included their interactions with peers, the work team, and patients–using the Flanagan's critical incident technique (CIT) [22]. This technique is considered valuable for understanding significant events related to professionals' behavior in well-defined situations. Critical incidents do not necessarily involve extreme gravity or life-threatening situations; instead, they encompass occurrences that are surprising, unexpected, or disturbing to the professional, prompting a certain level of analysis [22,23].

## Sampling and recruitment

From May 17th to October 11th, 2022, professors and third-year residents in the Family Medicine specialty from various federal entities within the Mexican Republic were invited via e-mail to participate in a Zoom session. Third-year residents were chosen for their tenure in the residency program, which exposed them to a wide range of incidents. The participating professors included associate and full professors with different years of teaching experience. During the sessions, the significance of the research was presented to sensitize, and motivate participants (rapport) [24,25]. We explained aspects of the CIT, communication problems, their effects, and the importance of addressing them. The study methodology was also described.

In these sessions, we emailed all attendees the informed consent form—in Word format—stating that participation was voluntary, anonymous, and would not affect their performance as resident or professors. The consent form detailed the benefits and risks of participating in the study and informed participants of their right to withdraw at any time.

Additionally, participants received Word documents containing questions about sociodemographic data and open-ended questions of critical incidents regarding teacher-resident communication challenges.

The elements that participants were required to address, regarding the critical incidents, are outlined in Table 1.

At the end of the sessions, attendees were given one week to submit their critical incidents and signed consent forms via e-mail. Only those who chose to participate freely returned the required documents in separate files.

Critical incident formats related to communication problems were included, while duplicate formats, poorly reported or incomplete, were excluded as they did not meet the descriptive elements mentioned in Table 1.

## Data collection and analysis

The research team comprised specialists in family medicine (I H-T, ON P-A), teaching experts (I H-T, LF R-H, G L-O), communication specialists (I H-T, ON P-A, LF R-H), and doctors in sciences, and education (G L-O, LF R-H). The obtained formats were independently analyzed according to the academic role (teacher or resident) to identify the most frequent communication problems.

**Table 1. Elements to be gathered using the CIT.**

| |
|---|
| a) Description of the critical incident centered on a communication problem during Family Medicine residency. |
| b) Context and circumstances in which the problem occurred. |
| c) Actions taken in response to the problem. |
| d) Participants' interpretation of the problem (analysis). |

Once the critical incidents that met the inclusion criteria were selected, the information was analyzed based on its content and contribution to each category of analysis. According to the methodology of Hughes [26], the following categories were previously established: organizational communication (related to documentation, information hierarchy, and institutional procedures), assertive communication (including sensitivity, empathy, problem-solving skills, or helpfulness), and effective communication (communication barriers). From the obtained data, an emergent category arose: asymmetric communication, which includes narratives about abuse of power, shouting, rudeness, and humiliating treatment.

To enhance the reliability of our findings and minimize individual biases, we employed investigator triangulation involving all authors. This approach leverages diverse perspectives to enrich the analysis and confirm findings [27]. Initially, each researcher independently reviewed the critical incident reports to confirm categories. We then convened to discuss our findings and resolve discrepancies through collaborative discussion. Subsequent reviews refined these categories and addressed any disagreements. Several meetings were held to resolve the remaining discrepancies and achieve consensus [28,29].

Saturation was reached when no new themes or subthemes were identified [30]; this occurred when 60% of the critical incidents were analyzed. However, the analysis was completed for all critical incidents that met the selection criteria.

## Results

Of those attending the Zoom session, 70 out of 103 professors (67.97%), and 50 out of 214 residents (23.36%) agreed to participate. A total of 224 critical incidents were collected (several participants reported more than one incident), 192 met the selection criteria (85.71%). Among these, 127 were reported by professors, of which 80 (63%) incidents were reported by women, and 47 (37%) by men. Residents reported 65 critical incidents, 32 (49.23%) described by women, and 33 (50.77%) by men. The average age for professors and residents was 42.44 years (±5.6), and 34.29 years (±5.01), respectively. Table 2 presents data on sex, age, years of teaching experience, and the number of critical incidents reported.

According to the critical incidents collected, four categories were confirmed: asymmetric communication, assertive communication, organizational communication, and effective communication. These categories include issues such as power dynamics, empathy, communication skills, poor internal communication, conflict resolution, among others. The frequency of these incidents was recorded for both professors and residents, highlighting the specific challenges identified by each group (Table 3).

### Asymmetric communication

Asymmetric communication refers to situations where there is an inequality in power, information, or influence among the parties involved in message transmission. This was prevalent in many identified critical incidents, negatively impacting patient care. Participants reported instances where their medical judgment was doubted by higher-ranking healthcare personnel, leading to delays in patient care. For example:

*"In an ongoing medical care shift, I presented a patient to the intermediate emergency area, and they refused to accept her because I was a resident. Despite explaining and justifying why the patient needed to be in that area, they refused to listen, and insisted that the attending physician had to give that indication. . . This significantly delayed patient care."* (Participant 28. Male resident)

**Table 2. Demographic characteristics and critical incidents reported by participants (N/A, not applicable).**

| Characteristic | | Full Professor | Associate Professor | Third-Year Resident |
|---|---|---|---|---|
| **Sex** | | | | |
| | Male | 19 | 8 | 24 |
| | Female | 41 | 2 | 26 |
| **Age (years)** | | | | |
| | 20–29 | 0 | 0 | 9 |
| | 30–39 | 20 | 4 | 30 |
| | 40–49 | 30 | 5 | 11 |
| | ≥50 | 10 | 1 | 0 |
| **Years of Teaching Experience** | | | | |
| | 1–3 | 30 | 4 | N/A |
| | 4–7 | 23 | 4 | N/A |
| | >7 | 7 | 2 | N/A |
| **Critical Incidents Reported—Men** | | | | |
| | n = 1 | 5 | 2 | 15 |
| | n = 2 | 14 | 6 | 9 |
| | **Total** | 33 | 14 | 33 |
| **Critical Incidents Reported—Women** | | | | |
| | n = 1 | 4 | 2 | 20 |
| | n = 2 | 37 | 0 | 6 |
| | **Total** | 78 | 2 | 32 |

Communication issues in this context even jeopardized patient safety, as reported by a resident:

> *"During my psychiatry rotation, I noticed that the treatments given by the attending physician were not consistent with the literature, and harming patients. . . I found a patient with signs of acute coronary ischemia, and three patients had hypertensive crises. I told the physician. . . he did not give the necessary attention because, for him, I was just a resident lacking experience."* (Participant 01. Female resident)

Asymmetric communication also impacted interactions among residents based on their rank. Inappropriate use of hierarchy justified abuses, creating uncertainty, fostering

**Table 3. Categories and subcategories of critical incidents according to academic figure.**

| Categories | Subcategories | Critical Incidents (Professors) | Critical Incidents (Residents) |
|---|---|---|---|
| **Asymmetric Communication** | • Power dynamics<br>• Degrading comments and belittlement<br>• Verbal aggression | 50 (39.37%) | 23 (35.38%) |
| **Assertive Communication** | • Message clarity and comprehension<br>• Empathy<br>• Communication skills<br>• Problem resolution | 45 (35.43%) | 10 (15.38%) |
| **Organizational Communication** | • Poor internal communication<br>• Misuse of communication channels<br>• Failure to communicate notices | 23 (18.11%) | 24 (36.92%) |
| **Effective Communication** | • Conflict resolution<br>• Communication barriers | 9 (7.08%) | 8 (12.3%) |
| **Total** | | **127 (100%)** | **65 (100%)** |

complicated interactions, and affecting the work environment. *"Verbal expressions of superiority by residents of higher grades affected group performance, and work environment."* (Participant 30. Female professor)

Within this category, critical incidents were identified where residents explicitly mentioned being subjected to degrading comments, belittlement, verbal abuse, shouting, mockery, rudeness, and intimidation by professors, attending physicians from other services, or residents of higher grades.

*"For instance, the professor blocked me from 'WhatsApp,' claiming, 'It's my phone, and I do what I want.' However, she gives instructions through that media... The physician not only has poor communication with some of us, but she is rude, and threatens to fail me... She laughs at me... I sent her several documents, and she refused to accept them, saying 'I do what I want'..."* (Participant 44. Female resident)

In describing these critical incidents, residents expressed feeling humiliated, belittled, and insecure, even expressing a *"desire to leave the residency program."* (Participant 17. Male resident)

## Assertive communication

Assertiveness plays a crucial role in building positive relationships in work environments. In critical incidents within this category, assertiveness was evident in active listening, empathy, and clarity in message delivery. These skills contributed to establishing channels of communication that facilitated the resolution of situations. *"Maintaining this openness prevented the development of situations that could have negatively impacted academic progress during residency"* (Participant 5. Male resident).

In critical incidents, the willingness of professors to engage in dialogue and resolve conflicts positively influenced residents' learning of clinical and communication skills. Assertiveness was crucial in physician-patient interactions, and in providing higher-quality medical services. One participant stated:

*"The physician reprimanded us in front of the patient and their family when we didn't know something, creating distrust in them... I approached him and talked about the doctor-patient relationship and how trust influenced it... and how these actions could harm the institution's image and the services we provided. The physician understood my point very well, and since that day, he has been correcting us privately."* (Participant 33. Female resident)

Assertive communication in critical incidents helped avoid misunderstandings caused by assumptions. Clearly describing ideas, active listening without prejudice, and efficient message reception were facilitated, promoting better performance:

*"I believe that knowing how to listen to the person approaching for support and waiting for them to talk about their issues makes communication flow. It allows us to assess what affects them and provide good guidance on decisions according to the problem."* (Participant 24. Female professor)

Within critical incidents narrated by professors, personal situations affecting the learning of some residents were mentioned. Active, sensitive, and empathetic listening skills helped professors seek joint solutions for the benefit of affected residents:

*"The residents were complaining about a colleague, saying they didn't want to be with her because she was too slow to work. After observing her, I noticed she had vision problems... I approached her to talk about how I could help her, and she said she didn't have money to buy glasses... I bought them for her and talked to some colleagues to form a study group to support her... After two months of having her glasses, the comments about her performance were positive."* (Participant 2. Male professor)

Detecting situations emotionally impacting students and being empathetic allowed professors *"to establish more direct communication to understand problems and seek personalized solutions."* (Participant 53. Female professor)

## Organizational communication

Organizational communication is crucial during medical residency since formats, notices, and administrative processes are part of daily life in healthcare services. Critical incidents expressed problems in communication for scheduling academic activities and aspects related to research.

Regarding the latter, several residents expressed difficulties in carrying out their thesis projects due to a need for clearer communication from authorities, and professors regarding institutional formats used to register the protocol with the ethics committees, progress delivery times, and thesis registration with relevant administrative bodies. Lack of communication, and the absence of institutional guidelines led to delays in developing their research projects, impacting knowledge dissemination. Some residents expressed organizational communication problems in this regard:

*"I was asked to present a research poster for a conference without prior information or specifications, and with less than 24 hours' notice; these generated multiple problems... the organization and communication were deficient... it seemed to me that they just wanted to fill spaces in that forum impromptu... it was a disaster in all aspects, and I gained no insights."* (Participant 11. Male resident)

Another common issue in critical incidents was the lack of communication regarding the scheduling of rotations, leading to conflicts, and affecting residents' learning:

*"A student was sent to a rotation in the endocrinology service but was rejected by the attending physician, claiming that he didn't know anything about the resident, the site, the coordinator, or anyone... The physician said that he was tired, and he didn't want more students... The resident lost her rotation in the endocrinology service."* (Participant 44. Female professor)

Lack of communication was a frequently recurring event, particularly during vacation and holiday periods. There was no information about the absence of professors, and residents resulted in conflicts affecting interpersonal relationships, as well as academic, and service activities.

*"I attended the class scheduled on the calendar... when I arrived, the classroom was empty, no students... I contacted the chief resident of the group, and he informed me that they did not attend because they went to a practice at a teaching center and 'forgot' to pass the memo."* (Participant 16. Male professor)

The lack of updating administrative procedures or changes in guidelines for specific processes was a common scenario. Some residents stated that neither professors nor attending physicians properly communicated procedures to patients and were not updated in rules and guidelines. This affected medical care, since miscommunicating the administrative requirements for referrals and service validity affected the authorization of medicines; *"this could lead to patients going home with incomplete or no doses of medications."* (Participant 21. Female resident)

## Effective communication

Effective communication involves the transmission of information in a clear manner, considering the needs of the interlocutor and their context. Some critical incidents highlighted the importance of feedback as an integral part of effective communication in medical education. Participants' narratives revealed problems caused by delivery relevant dates, and grades, particularly *"injustices in grading, even when assessment criteria had been communicated beforehand, leading to dissatisfaction and distance in the [teacher-student] relationship."* (Participant 19. Female professor)

In some critical incidents, the need for professors and residents to understand each other's perspectives and context was evident. The absence of these aspects led to misunderstandings and tensions in various scenarios. The importance of *keeping records and evidence of agreements to make communication more effective was mentioned; this helped prevent conflicts by providing a solid foundation for communication"* (Participant 58. Male professor). In some narratives, communication seemed forceful, and ineffective, resulting in unnecessary tensions.

*"Due to the absence of a teacher, residents had agreements to miss their rotations. . . I talked to them and informed that they had to comply the established schedule because, from the beginning, they accepted the residency and agreed to follow the schedules. The residents felt attacked, arguing that they were being asked for too much. . . Due to my communication, which was very energetic, they took it personally."* (Participant 62. Male professor)

In various critical incidents, it was observed that when regular communication channels were ineffective, they escalated to more tense, and confrontational situations with little chance of future success; this affected teaching work, and made the interaction between teachers and students less enjoyable, negatively impacting the residency environment:

*"A resident did not fulfill her academic responsibilities, despite verbal recommendations about the quality of her performance. . . when I did not see changes, I started drafting academic warnings. . . The student never told me she had some disabilities (hearing, visual, and motor). . . Months later, the student wrote a letter to the institutional authorities requesting a review of grades, arguing discrimination. . . as a professor, I never established an open communication channel where I asked for her opinion about the situation or if she had any problem preventing her from completing the specialty. . ."* (Participant 23. Female professor)

Effective communication was identified as being able to prevent further problems, even in situations that were harsh and unpleasant for the participants. Showing previous evidence of agreements, allowing those involved to communicate with respect, and presenting solid and reasonable arguments about the course of certain events meant acknowledging mistakes and the possibility of rectifying them, regardless of whether the participants liked it or not.

*"On one occasion, we had not submitted our theses on time. The professor, calmly, talked to us about the agreement we had made at the beginning of the school year, where we committed*

*to ourselves to hand in our theses according to the schedule. . . He showed us the evidence, a document with our signatures that made it clear that we knew when we had to submit our work. At first, we were defensive, but the professor explained to us the importance of agreements. . . Eventually, we understood that the responsibility was ours. . ."* (Participant 32. Female resident)

## Discussion

The communication problems analyzed describe incidents during family medicine residency. Asymmetric communication emerged as a critical element in the context of the teacher-student relationship. This dynamic was reflected in several incidents where communication problems based on power imbalances, and hierarchies. Hierarchy dynamics are a problem due to their lack of objectivity in communication by diminishing the importance of the message and focusing on situations that seek to demonstrate or ensure who can give orders and who must follow them, without considering the situation, and its context. This affects confidence in the residents, as well as their clinical judgment and their involvement in solving problems involving their professional training; it has been reported that, on various occasions, students may choose to remain silent because of their academic hierarchy, regardless of the relevance of their comments [31].

The existence of hierarchies in healthcare institutions is recognized as a triggering factor for unequal communication among professionals of different levels [15]. Disparities in power can lead to poor communication and generate a hostile work environment.

Medical residency is a comprehensive formative process in which students build and consolidate their ethical, and medical identity. The humiliation, belittlement, and verbal aggression mentioned by participants have a profound impact on dehumanizing doctors as professionals. In addition, it has been pointed out that the stress, to which students are subjected, negatively affects their learning and performance, contributing significantly to the emergence of mood disorders and suicidal ideation [32,33]. The described critical incidents align with findings from other studies indicating that the abuse of power in communication can erode trust, and collaboration in medical training environments [34–37].

Communication is essential in medical education, particularly in family medicine residency programs, as it plays a critical role in developing both clinical and interpersonal skills [5,38]. Our study identified several key areas where communication issues were prevalent.

Addressing these issues requires the implementation of structured communication training programs. These programs should be embedded into the medical curriculum and form part of residents' and professors' ongoing professional development. Moreover, it is essential for professors to engage in continuous education to improve their communication skills. Regular workshops and seminars focusing on communication strategies can significantly improve resident interactions, fostering a culture of respect and collaboration [39–41]. These educational efforts should include training in conflict resolution and feedback delivery, as our study identified these as major areas of concern.

It is noteworthy to highlight the importance of institutional support in creating clear and efficient communication channels, emphasizing empathy and mutual respect. Establishing standardized procedures for internal communication can prevent misunderstandings and ensure that important information is effectively disseminated.

Patient safety was significantly affected by problems of asymmetric communication. In such incidents, the figure of residents was nullified or minimized due to their academic status. Several studies have shown that open and effective communication among healthcare professionals is essential for making appropriate decisions, and preventing medical errors [42,43].

In the context of medical education, assertiveness is relevant since it is a skill that enables the establishment of positive relationships with colleagues and patients, facilitating the transmission of relevant medical information, whether technical or not. Moreover, it promotes management of difficult situations, informed decision making, managing patients, and enhancing teamwork performance in medical contexts [44,45]. Regarding the reported critical incidents, some narratives highlight the lack of clarity in the message, along with an absence of empathy. and poor communication skills for problem resolution, which affects learning and interaction due to the lack of trust generated in the residents.

Within the reported critical incidents, a variety of situations were identified where assertiveness played a crucial role, not only in solving issues related to the learning and training of medical specialists but also highlighting the need for assertiveness to address problems in medical education. Developing empathetic, assertive, and adequately communicative family physicians is imperative for the construction of the physician-patient relationship, as this relates to the ability to respect the patient and their family, fostering prosocial behaviors [46]. Assertiveness is a highly desirable skill in family physicians, as well as in other healthcare professionals.

As identified in critical incidents, assertive communication can prevent escalating situations that could lead to a "point of no return," where both, learning and well-being of teachers, students, and patients may be affected [47].

Another element identified in our results was problems in organizational communication. These were related to academic and service aspects. The lack of notice communication, clarity in scheduling activities during residency, and the misuse of institutional communication channels affected the development of research, teaching, and medical care activities. The loss of rotations across different services due to organizational communication problems implies incomplete training in various knowledge areas crucial in-patient care. It also affects the creation of potential networks with other complementary medical disciplines to family medicine [48]. In this regard, it has been noted that one of the main obstacles to be able to practice as competent, and professional physicians is the lack of acquisition of skills during their training. This is exacerbated in educational environments that prefer medical care over comprehensive professional training [13,49].

Examples of poor organizational communication in other medical and service contexts highlight the impact on processes, the dilution of responsibilities, the deterioration of medical care, an increase in medical complaints, a decrease in patient safety, and more significant expenditure of resources [50]. Some of these problems were shared in this research and caused friction within the residency. Therefore, addressing such communication problems is necessary to make institutional processes more efficient, which is crucial in medical education settings and in providing quality services [13,50,51].

The transmission of clear and direct information, as well as the creation of a environment of support and acceptance, is essential for establishing an effective dialogue where conflict resolution is vital for medical practice [52]. When analyzing critical incidents related to effective communication, the need to transmit information clearly, objectively, and in a personalized manner becomes evident. Several situations involving this type of communication were related to evaluation aspects, where residents expressed dissatisfaction with grading methods, and a lack of follow-up of their obligations in the residency, despite previous agreements. In the analyzed critical incidents, it was possible to identify that when communication was not effective, the deterioration in the teacher-student relationship was irrevocable. This affected not only interpersonal relationships but also the overall residency environment.

On the other hand, it was identified that effective communication was backed by physical evidence of previous agreements. This, even when there was resistance from some involved parties, prevented incidents from escalating to major confrontations, and encouraged the

assumption of responsibilities. In this regard, it is essential to note that communication is a skill valued by patients when evaluating the medical care received. In Mexico, it has been estimated that 3 out of 4 complaints against physicians, and health institutions are associated with ineffective communication [46].

This challenge is not unique to this country, which has led to proposals for developing structured training in communication skills that can facilitate patient-centered care [53]. Additionally, it has been identified that educational strategies in communication during medical residency not only improve residents' interactions with patients but also enhance the effectiveness with which they understand each other, illustrating the direct benefits of effective communication in mutual understanding [54]. These observations highlight the need for a reference on approaches to teaching effective communication skills, which are important to mitigating the adverse effects of poor communication during medical residency [55].

Parallel to the above, effective leadership qualities have been noted to promote open communication spaces, and a positive team dynamic to provide solutions, improve the ability to adapt to change, foster mutual respect, and cultivate a more favorable working environment for healthcare team members [56].

Communication is important in medical residencies and translates into patient-centered care and improved medical outcomes [13,57]. To address the communication problems identified in this study, an approach centered on open communication, and the creation of a culture of psychological safety is recommended. This would create an environment where professionals feel comfortable expressing their concerns, and opinions without repercussions [19,58,59].

It has been emphasized that effective communication skills should be included in the curricula of medical specialties, as communication efficiency and effectiveness can be improved through training [1,12,13,17]. However, compassionate, and empathetic bidirectional human communication between physician and patient is learned primarily from the actions of faculty. Higher-ranking physicians must be careful not to inhibit or affect communication channels that undermine the alterity of their peers, as these attitudes can quickly close such channels, eliminating any opportunity for learning [46]. Failure to promote communication skills in curricula, many negative communication behaviors, learned or emulated in medical training environments, may be perpetuated especially in medical residencies.

## Limitations

Due to the nature of critical incidents, where emphasis is placed on significant events that occurred in the past, there may have been recall biases among participants, as well as a lack of broader and more detailed description of incidents; this limitation could be addressed through other qualitative approaches [60]. Likewise, as the collectors of critical incidents were family medicine professors who have continuous contact with all participants, residents may have omitted information due to fears or inhibitions; this could explain their low participation (23.36%), potentially generating biases due to the nature of the applicator [61]. Additionally, the limited time available due to the educational and service demands of the residency, as well as the possibility that participants may have had other priorities, could have influenced their willingness to participate in this research, this same scenario could have occurred with professors.

## Conclusions

Communication problems impacted various family medicine residency scenarios, including teaching-learning processes, work environments, coexistence, and medical care. If this

scenario is not addressed, communication problems will continue to generate obstacles in the professional training of residents, which can negatively affect medical care and interactions with other healthcare professionals.

Research on the impact of communication during medical residencies is fundamental for restructuring and refining curricula. This allows for evaluating residents' performance in technical and diagnostic abilities and essential skills for their daily practice. Such evaluation is key to identifying and correcting asymmetries during medical training.

Medical education must incorporate content that addresses communication processes and the development of soft skills. These aspects can significantly improve specialists' empathy and social abilities, allowing them to interact more effectively with colleagues, patients, and their families.

Furthermore, it is important that the strengthening of communication skills continues beyond initial training. Continuing education must include updating and improvement programs in these areas, ensuring that physicians maintain and enhance their skills throughout their professional practice.

## Author Contributions

**Conceptualization:** Isaías Hernández-Torres.

**Data curation:** Luisa F. Romero-Henríquez, Geovani López-Ortiz.

**Formal analysis:** Isaías Hernández-Torres, Octavio N. Pons-Álvarez, Geovani López-Ortiz.

**Investigation:** Isaías Hernández-Torres, Octavio N. Pons-Álvarez, Luisa F. Romero-Henríquez, Geovani López-Ortiz.

**Methodology:** Isaías Hernández-Torres, Octavio N. Pons-Álvarez, Luisa F. Romero-Henríquez, Geovani López-Ortiz.

**Writing – original draft:** Isaías Hernández-Torres, Octavio N. Pons-Álvarez, Luisa F. Romero-Henríquez, Geovani López-Ortiz.

**Writing – review & editing:** Isaías Hernández-Torres, Luisa F. Romero-Henríquez, Geovani López-Ortiz.

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
