## [Decision Letter · Decision Letter 0]

16 May 2024

PONE-D-23-40801Difficulties in Teacher-Student Communication During Family Medicine Residency: A Qualitative StudyPLOS ONE

Dear Dr. López-Ortiz,

Thank you for submitting your manuscript to PLOS ONE. After careful consideration, we feel that it has merit but does not fully meet PLOS ONE’s publication criteria as it currently stands. Therefore, we invite you to submit a revised version of the manuscript that addresses the points raised during the review process.

We look forward to receiving your revised manuscript.

Kind regards,

Federica Canzan

Academic Editor

PLOS ONE

Journal Requirements:

Reviewers' comments:

Reviewer's Responses to Questions

**Comments to the Author**

1. Is the manuscript technically sound, and do the data support the conclusions?

Reviewer #1: Yes

Reviewer #2: Partly

2. Has the statistical analysis been performed appropriately and rigorously? 

Reviewer #1: Yes

Reviewer #2: N/A

3. Have the authors made all data underlying the findings in their manuscript fully available?

Reviewer #1: Yes

Reviewer #2: Yes

4. Is the manuscript presented in an intelligible fashion and written in standard English?

Reviewer #1: Yes

Reviewer #2: Yes

5. Review Comments to the Author

Reviewer #1: Thank you for inviting me to review this study, I want to begin by commending Geovani López-Ortiz and team on a fantastic effort on this paper. They have worked on an important area in medical education, specifically trainer-trainee communication, and they leverage strong methodology to reach promising results. Here are some recommendations that would improve their paper further:

Title and manuscript:

Perhaps, consider changing ‘difficulties’ to ‘challenges’, the latter might be a term that is better suited for this paper.

Abstract:

Overall perspective – well-written, concise and impactful

1. Deficiencies in communication among healthcare professionals have been acknowledged, with potential negative impacts on medical education, and various professional environments.

This is the opening sentence, seems a bit verbose, I would simplify this. Also, please clarify ‘acknowledged’ by whom, and what constitutes various professional environments.

2. “The main problems identified were abuse of power in communication, lack of communication skills, and the absence of institutional communication channels.” – this is actually a result, which does not appear in the results section and instead shows up for the first time in the conclusion. I would move this to the results section (which may need to be shortened) and make the conclusion as the overarching message/takeaway for your paper.

Introduction:

Overall – clear and articulate.

Major

Before adding the second sentence about lack of studies from Mexico, you should add details about existing studies globally. Use the funnel approach to first introduce global literature before moving more specifically to Mexico. Here are recommendations for studies you should cite as global literature on the topic.

• Myerholtz, L. (2014). Assessing family medicine residents' communication skills from the patient's perspective: evaluating the communication assessment tool. Journal of Graduate Medical Education, 6(3), 495-500.

• Merchant, A. A. H., Shaikh, N. Q., Afzal, N., Noorali, A. A., Abdul Rahim, K., Ahmad, R., ... & Haider, A. H. (2023). Disparities in patient-resident physician communication and counseling: A multi-perspective exploratory qualitative study. Plos one, 18(10), e0288549.

• Leung, K. K., Wang, W. D., & Chen, Y. Y. (2012). Multi-source evaluation of interpersonal and communication skills of family medicine residents. Advances in health sciences education, 17, 717-726.

Minor

- Medical residency is pivotal in “specialists” training – the phrase should be specialists’ training (apostrophe after s)

- “often sidelining the development of complementary skills” – can be better phrased as “often deprioritizing’. Sideline means there is absolutely no attention to it, deprioritize indicates that the amount of attention paid to it is reduced.

Methods

Overall – draws on practices from literature and used good methodological practices, certain areas need more elaboration.

Major

- The section on sampling needs more details. “From May 17th to October 11th, 2022, professors and third-year residents in the Family Medicine specialty from the Mexican Republic were invited via e-mail to a Zoom session.” Questions to answer: why were only third year residents chosen, which hospitals were included – single center or multi-center, were all of them full professors or range (assistant, associate, full)

- Consent: Was the consent acquired after the introductory session? It says that the consent was in writing – when and where was this administered. Was it informed consent? Were benefits and risks detailed, and participants clearly informed that they had a right to withdraw, and that this was voluntary participation.

- Exclusion criteria for critical incident forms: “while poorly reported, insufficient, incomplete, or duplicate formats were excluded.” How many were excluded? Who determined that certain forms were “insufficient”?

- “Discrepancies were resolved through multiple rounds of discussion using researcher triangulation.” Elaborate on this further – how were these discussions on resolution of disparities structured – what was the methodology?

Minor

- This is not a complete sentence “Qualitative study, the present research was app..”

- Research team: The research team comprised specialists in family medicine, teaching experts, communication specialists, and doctors in sciences, and education. Please be more specific – you can add the initials of the authors to clarify which author brought what level of expertise to the table.

Results:

Overall – great attempt, but there are some deficiencies and areas of overlap between subcategories.

Major:

- “50 out of 214 residents 23.36% agreed to participate” – this needs some reflection in the limitations section under discussion. What could be reasons why >3/4th of the residents did not agree to participate? This has been reported later, please see my note below in the limitations section.

- “assertiveness was evident in active listening, empathy, and clarity in message delivery” are you sure if assertiveness is the right word that refers to these attributes, specifically empathy? Assertiveness is more about the level of confidence (https://dictionary.cambridge.org/us/dictionary/english/assertive) , which relates better with - “clarity in message delivery” but not the other two.

The section on organizational communication (theme) has been written well, explains different components and perspectives with supporting quotations. Well done.

- For the section on “effective communication”, there is strong overlap between the subcategory “message understanding” and the subcategory for theme 2 which is “Clarity in the message”. Consider combining the two.

Minor:

“80 (63%) incidents were reported by women, and 47 (37%) by men” – gender differences in the number of critical incidents reported. Given this, you should highlight which comments came from which gender. Add the gender of the participant in the brackets when you mention the participant number.

Discussion:

Overall - Well-articulated, starts strong.

Major:

“In Mexico, it has been estimated that 3 out of 4 complaints against physicians, and health institutions are associated with ineffective communication [36].”

I would also add some global literature where you refer to effective/ineffective communication. Here are some citations to include:

- Turner, J. W., Robinson, J., Morris, E., Oberkircher, K., Rios, R., & Roett, M. (2020). Resident reflections on resident-patient communication during family medicine clinic visits. Patient education and counseling, 103(3), 484–490. https://doi.org/10.1016/j.pec.2019.09.011

- Shahbaz, H., Noorali, A. A., Inam, M., Qadeer, N., Merchant, A. A. H., Khan, A. A., Afzal, N., Abdul Rahim, K., Munaf, I., Ahmad, R., Tariq, M., & Haider, A. H. (2022). Developing a communication-skills training curriculum for resident-physicians to enhance patient outcomes at an academic medical centre: an ongoing mixed-methods study protocol. BMJ open, 12(8), e056840. https://doi.org/10.1136/bmjopen-2021-056840

- Jansen, K. L., & Rosenbaum, M. E. (2016). The State of Communication Education in Family Medicine Residencies. Family medicine, 48(6), 445–451.

Limitations and missing demographics table:

- There is mention of one possible reason for the low resident turnout “may have omitted information due to fears or inhibitions” – do you think this was the sole reason – that the data collectors were professors, leading to fear. What could be other reasons? Additionally, do you think that the data collectors being the residents’ own professors could have impacted the responses of those who did participate? Maybe they were also hesitant? Comment on this.

- Were there any gender-based differences? More women reported incidents, but is that because of a higher tendency to report, or because of a higher number of women in the study sample. If it is the former, what are some reasons why – these could be explored in the discussion.

- A table of overall participant demographics needs to be added. You can include gender/biological sex, age, level of training/professor-ship, how many reported 1 incident and how many >1.

- I would also be interested in seeing if there are any differences in those that agreed to respond and those who did not respond – is there a selection bias there? For example: out of all women who were sent the email, a higher proportion responded, compared to all men who were sent the email. This difference in response/no response does not need to be added to the paper, but I would like to review it in your response letter, to see if there was any selection bias, which would then need to be added as a limitation.

Conclusion: Very brief, I would add a couple of sentences on takeaway thoughts and overarching picture/perspective. What is the key message and big picture that you want the readers to leave with? Additionally, what are some directions for future studies.

Overall, your team has done a great job and I would be happy to support this manuscript further pending revisions. Best of luck!

Reviewer #2: Thank you for the opportunity to revise this interesting manuscript that talk about a never-ending issue in clinical practice.

However, some information are missing to fully understand the methods used.

Introduction

Reference to Mexico is not contextualized and the reader could have difficulties to understand why you are citing the country

Communication is explored under several perspectives (with patients, with the team, with the teachers) and is not clear which one is relevant for the study

Differences among the terms students, teachers and residents could not be clear to the reader, needing of specify what do you mean

Previous research findings on this topic are missing

Gap in research is missing

The aim of the study could be more defined

Methods

Why did you choose to investigate the phenomenon from the perspective of critical incidents?

How did you recruit participants?

Inclusion criteria?

From the sentence “Participants received Word-format…”, you could move the paragraph under data collection

What sociodemographic data did you collect? Did you collect work experience data?

How should participants report information? Were the answers provided as open questions? I’m concerned about differences in reporting among participants. You could have missed important data on events.

What did you mean with “poorly reported/insufficient/incomplete etc?” which were the criteria?

How did you evaluate if the critical incidents reported by participants was pertinent to your research question? Which criteria did you adopted?

How did you analyze the data? Content analysis? Thematic analysis? The method of data analysis could be better described.

Given the missing information in the method section, I have some difficulties to judge the results and discussion at the moment.

However, these sections seem written well and are understandable. I suggest removing “theme” from the title of the subheadings.

In the discussion section, more emphasis should be given to training on communication skills among the team and other strategies to prevent communication issues between teachers and residents, such as training and continuing education for teachers.

I hope that suggestions could help to improve the manuscript.

6. PLOS authors have the option to publish the peer review history of their article (what does this mean?). If published, this will include your full peer review and any attached files.

Reviewer #1: No

Reviewer #2: No

---

## [Author Response · Author response to Decision Letter 0]

5 Jul 2024

Comments to the Author

Reviewer #1: Thank you for inviting me to review this study, I want to begin by commending Geovani López-Ortiz and team on a fantastic effort on this paper. They have worked on an important area in medical education, specifically trainer-trainee communication, and they leverage strong methodology to reach promising results. Here are some recommendations that would improve their paper further:

Dear reviewer,

We sincerely appreciate the time you have taken to review this manuscript and your kind comments regarding our work. Your suggestions and opinions are invaluable to us. We have carefully considered your recommendations and implemented the following modifications to improve the clarity and quality of our article.

We hope these changes address your concerns and enhance the overall quality of our manuscript. 

All changes and modifications made to this work can be verified in the Revised Manuscript with Track Changes.

Title and manuscript:

Perhaps, consider changing ‘difficulties’ to ‘challenges’, the latter might be a term that is better suited for this paper.

We appreciate the suggestion and agree with the reviewer that ‘challenges’ is a more appropriate term for the title. We have made the corresponding changes in the titles and the objective to ensure consistency throughout the study.

Abstract:

Overall perspective – well-written, concise and impactful

Thank you for the comment.

1. Deficiencies in communication among healthcare professionals have been acknowledged, with potential negative impacts on medical education, and various professional environments.

This is the opening sentence, seems a bit verbose, I would simplify this. Also, please clarify ‘acknowledged’ by whom, and what constitutes various professional environments.

Thank you for your comment. In response, we have made the following modifications:

Deficiencies in communication among healthcare professionals, recognized by medical educators and healthcare institutions, can negatively impact medical education and clinical practice.

The phrase was simplified to deliver a more direct message. Additionally, we clarified what we meant by professional environments.

2. “The main problems identified were abuse of power in communication, lack of communication skills, and the absence of institutional communication channels.” – this is actually a result, which does not appear in the results section and instead shows up for the first time in the conclusion. I would move this to the results section (which may need to be shortened) and make the conclusion as the overarching message/takeaway for your paper.

We appreciate your suggestion and agree that the main problems identified, as referenced in the conclusion, are elements of the results. In response, we have made the following modifications:

“… and 4) effective communication. The main problems reported were abuse of power in communication, lack of communication skills, and the absence of institutional communication channels. These issues significantly impacted learning, work environment, interpersonal relationships, and medical care.

The conclusion has been restructured as follows:

This study highlights communication issues within Family Medicine residency in Mexico. The issues detected hindered learning and effective collaboration and negatively impacted the work environment, interpersonal relationships, and the quality of medical care. These findings underscore the urgent need to reorient the medical specialty curriculum towards an approach that includes communication skills.

Introduction:

Overall – clear and articulate.

Thank you for the comment.

Major

Before adding the second sentence about lack of studies from Mexico, you should add details about existing studies globally. Use the funnel approach to first introduce global literature before moving more specifically to Mexico. Here are recommendations for studies you should cite as global literature on the topic.

• Myerholtz, L. (2014). Assessing family medicine residents' communication skills from the patient's perspective: evaluating the communication assessment tool. Journal of Graduate Medical Education, 6(3), 495-500.

• Merchant, A. A. H., Shaikh, N. Q., Afzal, N., Noorali, A. A., Abdul Rahim, K., Ahmad, R., ... & Haider, A. H. (2023). Disparities in patient-resident physician communication and counseling: A multi-perspective exploratory qualitative study. Plos one, 18(10), e0288549.

• Leung, K. K., Wang, W. D., & Chen, Y. Y. (2012). Multi-source evaluation of interpersonal and communication skills of family medicine residents. Advances in health sciences education, 17, 717-726.

We appreciate the comment. Indeed, referring to Mexico without prior context can seem too abrupt when developing the topic. To give the introduction more depth, we have added the following, including the recommended citations as well as two additional ones due to their relevance to this study:

… skills such as communication [1]. Several studies have shown that deficiencies in communication skills among family medicine residents and specialists are a systemic problem caused by inadequate training and time constraints, which negatively impact interactions in academic and service environments [2-6]. Evaluating these skills comprehensively allows for the identification of areas for improvement [4], biases and disparities in interaction environments [5], and the need to implement specific cultural strategies to address these deficiencies [6]. Therefore, the need to incorporate practical activities for teaching effective communication strategies has been identified in various environments where residency takes place [5].

Additionally, we have made the following modifications at the request of reviewer 2:

Despite the importance and recognition of developing these skills in different parts of the world, communication problems are common in Mexico, where cultural and educational frameworks have historically emphasized technical and clinical skills over communication abilities. As a result, content related to communication in various family medicine curricula in the country is not sufficiently explored or nonexistent [7-10].

Minor

- Medical residency is pivotal in “specialists” training – the phrase should be specialists’ training (apostrophe after s)

- “often sidelining the development of complementary skills” – can be better phrased as “often deprioritizing’. Sideline means there is absolutely no attention to it, deprioritize indicates that the amount of attention paid to it is reduced.

Thank you for highlighting these aspects. The suggested changes have been implemented and can be seen in the Revised Manuscript with Track Changes.

Methods

Overall – draws on practices from literature and used good methodological practices, certain areas need more elaboration.

We appreciate the comment. The suggested changes have been addressed throughout the Methods section.

Major

- The section on sampling needs more details. “From May 17th to October 11th, 2022, professors and third-year residents in the Family Medicine specialty from the Mexican Republic were invited via e-mail to a Zoom session.” Questions to answer: why were only third year residents chosen, which hospitals were included – single center or multi-center, were all of them full professors or range (assistant, associate, full)

Thank you for your insightful comments regarding the Sampling and Recruitment section. We appreciate the opportunity to clarify and provide additional details. In response, we have made the following clarifications:

From May 17th to October 11th, 2022, professors and third-year residents in the Family Medicine specialty from various federal entities within the Mexican Republic were invited via e-mail to participate in a Zoom session. Third-year residents were chosen for their tenure in the residency program, which exposed them to a wide range of incidents. The participating professors included associate and full professors with different years of teaching experience.

Additionally, in the “Study Design and Setting” section, we clarified that this is a multicenter qualitative study.

- Consent: Was the consent acquired after the introductory session? It says that the consent was in writing – when and where was this administered. Was it informed consent? Were benefits and risks detailed, and participants clearly informed that they had a right to withdraw, and that this was voluntary participation.

Thank you for your questions regarding the consent process. We appreciate the opportunity to clarify this information. We described the following in the manuscript.

In these sessions, we emailed all attendees the informed consent form—in Word format—stating that participation was voluntary, anonymous, and would not affect their performance as students or professors. The consent form detailed the benefits and risks of participating in the study and informed participants of their right to withdraw at any time. 

Additionally, participants received Word documents containing questions about sociodemographic data and descriptions of critical incidents regarding teacher-resident communication challenges.

Furthermore, we added: At the end of the sessions, attendees were given one week to submit their critical incidents and signed consent forms via e-mail. Only those who chose to participate freely returned the required documents in separate files.

- Exclusion criteria for critical incident forms: “while poorly reported, insufficient, incomplete, or duplicate formats were excluded.” How many were excluded? Who determined that certain forms were “insufficient”?

We appreciate these observations. Regarding the excluded forms, we mentioned in the Results section, first paragraph, second, and third lines that out of 224 incidents collected, 192 (85.71%) met the selection criteria.

We have removed the term “insufficient,” as we consider it not the most appropriate word and overlaps with “incomplete.”

Following Hughes’ methodology, the authors, after a round of discussions, decided to exclude those incidents that did not meet the criteria contained in Table 1, namely those that did not report communication problems, those that described incidents only partially (e.g., not providing the context and circumstances), and those that described incidents in a very brief manner (e.g., a professor raised his voice during a shift; without further details).

- “Discrepancies were resolved through multiple rounds of discussion using researcher triangulation.” Elaborate on this further – how were these discussions on resolution of disparities structured – what was the methodology?

Thank you for the comment and for the opportunity to clarify this section further. We make the following clarifications indicated in the methodology.

To enhance the reliability of our findings and minimize individual biases, we employed investigator triangulation involving all authors. This approach leverages diverse perspectives to enrich the analysis and confirm findings [27]. Initially, each researcher independently reviewed the critical incident reports to confirm categories. We then convened to discuss our findings and resolve discrepancies through collaborative discussion. Subsequent reviews refined these categories and addressed any disagreements. Several meetings were held to resolve the remaining discrepancies and achieve consensus [28,29].

Minor

- This is not a complete sentence “Qualitative study, the present research was app..”

We apologize for the oversight. We have clarified that this was a multicenter qualitative study involving Family Medicine professors and residents from the Mexican Republic.

- Research team: The research team comprised specialists in family medicine, teaching experts, communication specialists, and doctors in sciences, and education. Please be more specific – you can add the initials of the authors to clarify which author brought what level of expertise to the table.

We appreciate the comment. The participants' initials who contributed to the different disciplines have been added.

Results:

Overall – great attempt, but there are some deficiencies and areas of overlap between subcategories.

Thank you for your constructive feedback and for acknowledging the effort put into this study. We appreciate your comments regarding the deficiencies and areas of overlap between subcategories.

We have carefully reviewed the subcategories to address these concerns and have made the necessary adjustments to clarify and distinguish them more effectively. We believe these revisions enhance the clarity and precision of our findings.

Major:

- “50 out of 214 residents 23.36% agreed to participate” – this needs some reflection in the limitations section under discussion. What could be reasons why >3/4th of the residents did not agree to participate? This has been reported later, please see my note below in the limitations section.

In the limitations section, we addressed this observation.

- “assertiveness was evident in active listening, empathy, and clarity in message delivery” are you sure if assertiveness is the right word that refers to these attributes, specifically empathy? Assertiveness is more about the level of confidence (https://dictionary.cambridge.org/us/dictionary/english/assertive) , which relates better with - “clarity in message delivery” but not the other two.

Thank you for your comment. While assertiveness is traditionally associated with confidence and clarity in communication (as defined by the Cambridge Dictionary), it also encompasses understanding and respecting the rights and feelings of others. This broader view of assertiveness includes qualities like empathy and active listening, which facilitate respectful and effective interactions. Therefore, using “assertiveness” in this context highlights its role as an integrative element in communication that includes empathy as a key component (10.15649/2346030X.757).

The section on organizational communication (theme) has been written well, explains different components and perspectives with supporting quotations. Well done.

Thank you very much for your comment.

- For the section on “effective communication”, there is strong overlap between the subcategory “message understanding” and the subcategory for theme 2 which is “Clarity in the message”. Consider combining the two.

We appreciate your suggestion to combine the subcategory “message understanding” with “message clarity” from theme 2. After thorough consideration, we recognize the suggestion and have decided to combine these subcategories into a single entity. This new combined subcategory, “Message Clarity and Comprehension,” will address both the sender’s ability to convey information clearly and the recipient’s ability to comprehend it.

Minor:

“80 (63%) incidents were reported by women, and 47 (37%) by men” – gender differences in the number of critical incidents reported. Given this, you should highlight which comments came from which gender. Add the gender of the participant in the brackets when you mention the participant number.

We appreciate the comment and consider it pertinent, given the gender distribution. We have included the gender in each comment.

Discussion:

Overall - Well-articulated, starts strong.

Thank you very much for your comment.

Major:

“In Mexico, it has been estimated that 3 out of 4 complaints against physicians, and health institutions are associated with ineffective communication [36].”

I would also add some global literature where you refer to effective/ineffective communication. Here are some citations to include:

- Turner, J. W., Robinson, J., Morris, E., Oberkircher, K., Rios, R., & Roett, M. (2020). Resident reflections on resident-patient communication during family medicine clinic visits. Patient education and counseling, 103(3), 484–490. https://doi.org/10.1016/j.pec.2019.09.011

- Shahbaz, H., Noorali, A. A., Inam, M., Qadeer, N., Merchant, A. A. H., Khan, A. A., Afzal, N., Abdul Rahim, K., Munaf, I., Ahmad, R., Tariq, M., & Haider, A. H. (2022). Developing a communication-skills training curriculum for resident-physicians to enhance patient outcomes at an academic medical centre: an ongoing mixed-methods study protocol. BMJ open, 12(8), e056840. htt

---

## [Decision Letter · Decision Letter 1]

12 Aug 2024

PONE-D-23-40801R1Challenges in Teacher-Student Communication During Family Medicine Residency: A Qualitative StudyPLOS ONE

Dear Dr. López-Ortiz,

Thank you for submitting your manuscript to PLOS ONE. After careful consideration, we feel that it has merit but does not fully meet PLOS ONE’s publication criteria as it currently stands. Therefore, we invite you to submit a revised version of the manuscript that addresses the points raised during the review process.

We look forward to receiving your revised manuscript.

Kind regards,

Federica Canzan

Academic Editor

PLOS ONE

Journal Requirements:

Reviewers' comments:

Reviewer's Responses to Questions

**Comments to the Author**

1. If the authors have adequately addressed your comments raised in a previous round of review and you feel that this manuscript is now acceptable for publication, you may indicate that here to bypass the “Comments to the Author” section, enter your conflict of interest statement in the “Confidential to Editor” section, and submit your "Accept" recommendation.

Reviewer #1: All comments have been addressed

2. Is the manuscript technically sound, and do the data support the conclusions?

Reviewer #1: Yes

3. Has the statistical analysis been performed appropriately and rigorously? 

Reviewer #1: Yes

4. Have the authors made all data underlying the findings in their manuscript fully available?

Reviewer #1: Yes

5. Is the manuscript presented in an intelligible fashion and written in standard English?

Reviewer #1: Yes

6. Review Comments to the Author

Reviewer #1: Thank you for reviewing all recommendations/suggestions and incorporating most of them. I also appreciate the clarifications where you did not incorporate suggestions - such as the response on why you felt using assertiveness was appropriate.

The overall manuscript looks in excellent shape now and is close to publication. I am submitting a recommendation to accept the manuscript rather than a minor revision, so that it saves you time and another revision is not required. However, there were two comments that were not addressed, and the response letter stated that these changes have been incorporated.

The first one is a grammatical improvement: https://www.merriam-webster.com/grammar/what-happens-to-names-when-we-make-them-plural-or-possessive

The second one is just a change of phrasing to make it more palatable.

You are welcome to incorporate these changes or retain the original, but I would recommend making sure that the grammar sounds right (in the first one).

Other than that, great work and best of luck for getting this important work published soon!

_____

Copied below from response letter - I did not find these changes in the revised version/track changes.

- Medical residency is pivotal in “specialists” training – the phrase should be specialists’ training (apostrophe after s)

- “often sidelining the development of complementary skills” – can be better phrased as “often deprioritizing’. Sideline means there is absolutely no attention to it, deprioritize indicates that the amount of attention paid to it is reduced.

Thank you for highlighting these aspects. The suggested changes have been implemented and can be seen in the Revised Manuscript with Track Changes.

7. PLOS authors have the option to publish the peer review history of their article (what does this mean?). If published, this will include your full peer review and any attached files.

Reviewer #1: No

---

## [Author Response · Author response to Decision Letter 1]

14 Aug 2024

Comments to the Author

Reviewer #1: Thank you for reviewing all recommendations/suggestions and incorporating most of them. I also appreciate the clarifications where you did not incorporate suggestions - such as the response on why you felt using assertiveness was appropriate.

The overall manuscript looks in excellent shape now and is close to publication. I am submitting a recommendation to accept the manuscript rather than a minor revision, so that it saves you time and another revision is not required. However, there were two comments that were not addressed, and the response letter stated that these changes have been incorporated.

The first one is a grammatical improvement: https://www.merriam-webster.com/grammar/what-happens-to-names-when-we-make-them-plural-or-possessive

The second one is just a change of phrasing to make it more palatable.

You are welcome to incorporate these changes or retain the original, but I would recommend making sure that the grammar sounds right (in the first one).

Other than that, great work and best of luck for getting this important work published soon!

Copied below from response letter - I did not find these changes in the revised version/track changes.

- Medical residency is pivotal in “specialists” training – the phrase should be specialists’ training (apostrophe after s)

- “often sidelining the development of complementary skills” – can be better phrased as “often deprioritizing’. Sideline means there is absolutely no attention to it, deprioritize indicates that the amount of attention paid to it is reduced.

Dear reviewer 

Thank you very much for your positive feedback and for recommending our manuscript for acceptance. We greatly appreciate your thorough review and the valuable suggestions you have provided throughout this process.

Regarding the two comments you mentioned that were not addressed in our previous revision:

Grammatical improvement: We sincerely apologize for not incorporating the correction earlier. We have now adjusted the phrase “Medical residency is pivotal in specialists’ training” by correctly placing the apostrophe after the "s" in "specialists’" to ensure grammatical accuracy.

Change of phrasing: We also regret that the phrasing change was not included in the previous version. We have now revised the sentence “often sidelining the development of complementary skills” to “often deprioritizing the development of complementary skills,” as per your recommendation, to better convey the intended meaning.

We have made these changes in the revised manuscript, and we believe that they further improve the clarity and quality of the work.

Thank you once again for your time and feedback.

---

## [Editor Report · Decision Letter 2]

1 Sep 2024

Challenges in Teacher-Student Communication During Family Medicine Residency: A Qualitative Study

PONE-D-23-40801R2

Dear Dr. López-Ortiz,

We’re pleased to inform you that your manuscript has been judged scientifically suitable for publication and will be formally accepted for publication once it meets all outstanding technical requirements.

Kind regards,

Federica Canzan

Academic Editor

PLOS ONE
---

## [Editor Report · Acceptance letter]

6 Sep 2024

PONE-D-23-40801R2 

PLOS ONE

Dear Dr. López-Ortiz, 

I'm pleased to inform you that your manuscript has been deemed suitable for publication in PLOS ONE. Congratulations! Your manuscript is now being handed over to our production team.

Kind regards, 

on behalf of

Professor Federica Canzan 

Academic Editor

PLOS ONE